# A functional screen of RNA binding proteins identifies genes that promote or limit the accumulation of CD138+ plasma cells

David J Turner[1,2], Alexander Saveliev[1], Fiamma Salerno[1], Louise S Matheson[1], Michael Screen[1], Hannah Lawson[3], David Wotherspoon[3], Kamil R Kranc[3], Martin Turner[1]*

[1]Immunology Programme, The Babraham Institute,Babraham Research Campus, Cambridge, United Kingdom; [2]RNA Biology Laboratory, Center for Cancer Research, National Cancer Institute (NCI), Frederick, United States; [3]Laboratory of Haematopoietic Stem Cell and Leukaemia Biology, Queen Mary University of London, London, United Kingdom

**Abstract** To identify roles of RNA binding proteins (RBPs) in the differentiation or survival of antibody secreting plasma cells we performed a CRISPR/Cas9 knockout screen of 1213 mouse RBPs for their ability to affect proliferation and/or survival, and the abundance of differentiated CD138 + cells in vitro. We validated the binding partners CSDE1 and STRAP as well as the m⁶A binding protein YTHDF2 as promoting the accumulation of CD138 + cells in vitro. We validated the EIF3 subunits EIF3K and EIF3L and components of the CCR4-NOT complex as inhibitors of CD138 + cell accumulation in vitro. In chimeric mouse models YTHDF2-deficient plasma cells failed to accumulate.

## Editor's evaluation

This paper utilizes an elegant Crispr-Cas9 screen to identify RNA binding proteins that may regulate B cell differentiation. With some additional work to verify that the identified proteins are important in vivo, the paper will be of interest to a broad audience of immunologists studying the signals regulating B cell differentiation during an immune response.

## Introduction

Humoral immunity requires the generation of antibody-secreting plasmablasts and plasma cells. These are derived from naive B cell precursors that enter the extrafollicular response and the germinal centre (GC) reaction (*Nutt et al., 2015*). Germinal centre derived antibody secreting cells may migrate to the bone marrow where they survive as long-lived plasma cells secreting high-affinity antibody.

The B- to plasma-cell transition requires substantial reprogramming of the transcriptome. This is achieved in part by the actions of transcription factors and epigenetic regulators that act within a layered system to control the timing of differentiation and its coordination with extracellular cues. Dynamic control within this system is further endowed by the integration of signal transduction pathways and post-transcriptional regulators. A central role for post-transcriptional control is mediated by microRNAs such as *Mir148a* (*Porstner et al., 2015*) and *Mir155* (*Lu et al., 2014*) which promote and *Mir125b* (*Gururajan et al., 2010*) which suppresses plasma cell differentiation. RNA binding proteins (RBPs) act co- or post-transcriptionally to influence the quality and quantity of expressed genes.

*For correspondence:
martin.turner@babraham.ac.uk

**Competing interest:** The authors declare that no competing interests exist.

However, while estimates of the numbers of RBPs encoded in the mammalian genome vary between one to five thousand, few have been implicated in plasma cell differentiation, survival, and migration. These include, HNRNPLL which promotes differentiation indirectly by suppressing BCL6 expression (*Chang et al., 2015*); the non-canonical poly(A) polymerase FAM46C/TENT5C which acts as a tumour suppressor in myeloma (*Bilska et al., 2020*); and ZFP36L1 which is required for the emigration of plasma cells (*Saveliev et al., 2021*).

Motivated by a lack of known roles of RBPs in plasma cell emergence and survival, we developed a CRISPR library targeting mouse RBPs and performed genetic screens for roles in B cell proliferation and survival, and for roles in regulating the accumulation of CD138 + cells in vitro as a surrogate for authentic plasma cells. We identified 103 RBPs that promote and 189 RBPs that inhibit plasma cell abundance and validated some of these using follow up assays with independent single guide RNA (sgRNA) sequences. Using mouse models, we demonstrated that YTHDF2, an N6-methyladenosine (m⁶A) RNA modification binding protein, acting within B cells is necessary for the accumulation of plasma cells in the bone marrow.

## Results and discussion

### Genetic screening identifies RBPs regulating CD138+ cell abundance

We curated a list of 1213 mouse RBPs (excluding ribosomal subunit proteins) compiled from the compendium by Gerstberger et al and presence in at least two RNA interactome capture studies (*Castello et al., 2012*; *Baltz et al., 2012*; *Beckmann et al., 2015*; *Gerstberger et al., 2014*; *Kwon et al., 2013*; *He et al., 2016*; *Liepelt et al., 2016*; *Supplementary file 1A*). A custom sgRNA library targeting these was constructed in a vector backbone expressing puromycin resistance and CD90.1 as selection markers (*Figure 1—figure supplement 1A*). The library encodes 10 sgRNAs per gene, targets 72 positive control genes and contains 500 negative control sgRNAs. It contains 13,350 sgRNAs in total with >98.8% of sgRNAs normally distributed within a 16-fold range (*Figure 1—figure supplement 1B*; *Supplementary file 1B*).

To screen RBPs that regulate cell expansion (proliferation or survival), primary mouse B cells expressing Cas9 from the *Rosa26* locus were cultured in vitro on fibroblasts expressing CD40ligand and BAFF for 4 days with IL-4 followed by 4 days with IL-21[15]. The representation of sgRNAs at day 4 and day 8 was determined by next generation sequencing (NGS) and compared (*Figure 1—figure supplement 1C*; *Supplementary file 1C*). sgRNAs targeting *Trp53* and *Myc* enriched as expected for inhibiting and supporting cell expansion, respectively (*Figure 1—figure supplement 1D, E*). *Rc3h1*, *Caprin1*, and *Fam46c* were the only RBPs enriched for limiting B cell proliferation or survival (*Figure 1—figure supplement 1D, E*). FAM46C has been implicated in impairing the proliferation and/or survival of B cells (*Mroczek et al., 2017*; *Bilska et al., 2020*). Further supporting the predictive power of our dataset, we validated the role for *Rc3h1* as limiting proliferation or survival with individual sgRNAs (*Figure 1—figure supplement 1F*).

On day 8 of the culture, we sorted cells based on the expression of CD138, a marker of differentiated plasma cells that co-express CD267, IRF4, and BLIMP1 (encoded by *Prdm1*) (*Figure 1—figure supplement 1G*), and determined the representation of sgRNAs by NGS (*Figure 1—figure supplement 1C*; *Supplementary file 1D*). To identify RBPs that regulate the accumulation of CD138 + cells, but not B cell expansion between days 4 and 8, we considered the intersection between the two screens (*Figure 1A*). The transcription factors: *Bach2*, *Bcl6*, *Spi1*, and *Irf8* enriched for inhibiting CD138 + cell accumulation, and *Prdm1* enriched for promoting CD138 + cell accumulation, while *Irf4* was required for CD138 + cell accumulation as well as B cell expansion (*Figure 1A, B*). These results are consistent with the established roles (*Bilska et al., 2020*; *Ochiai et al., 2008*; *Ochiai et al., 2006*; *Muto et al., 2010*; *Shaffer et al., 2000*; *Tunyaplin et al., 2004*; *Carotta et al., 2014*; *Shapiro-Shelef et al., 2003*; *Lin et al., 2002*; *Shaffer et al., 2002*; *Sciammas et al., 2006*; *Ochiai et al., 2013*) of these genes in B cells and plasma cells, and therefore underpin our confidence that novel findings in this system reflect meaningful biological effects.

We focused on the 137 RBPs that enriched for a role in regulating plasma cell accumulation that did not enrich for a role in the proliferation and/or survival of B cells (*Supplementary file 1E*). These enriched genes may regulate the emergence, proliferation or survival of CD138 + cells. We validated CSDE1 and STRAP which bind to each other (*Hunt et al., 1999*) as promoting CD138 + cell

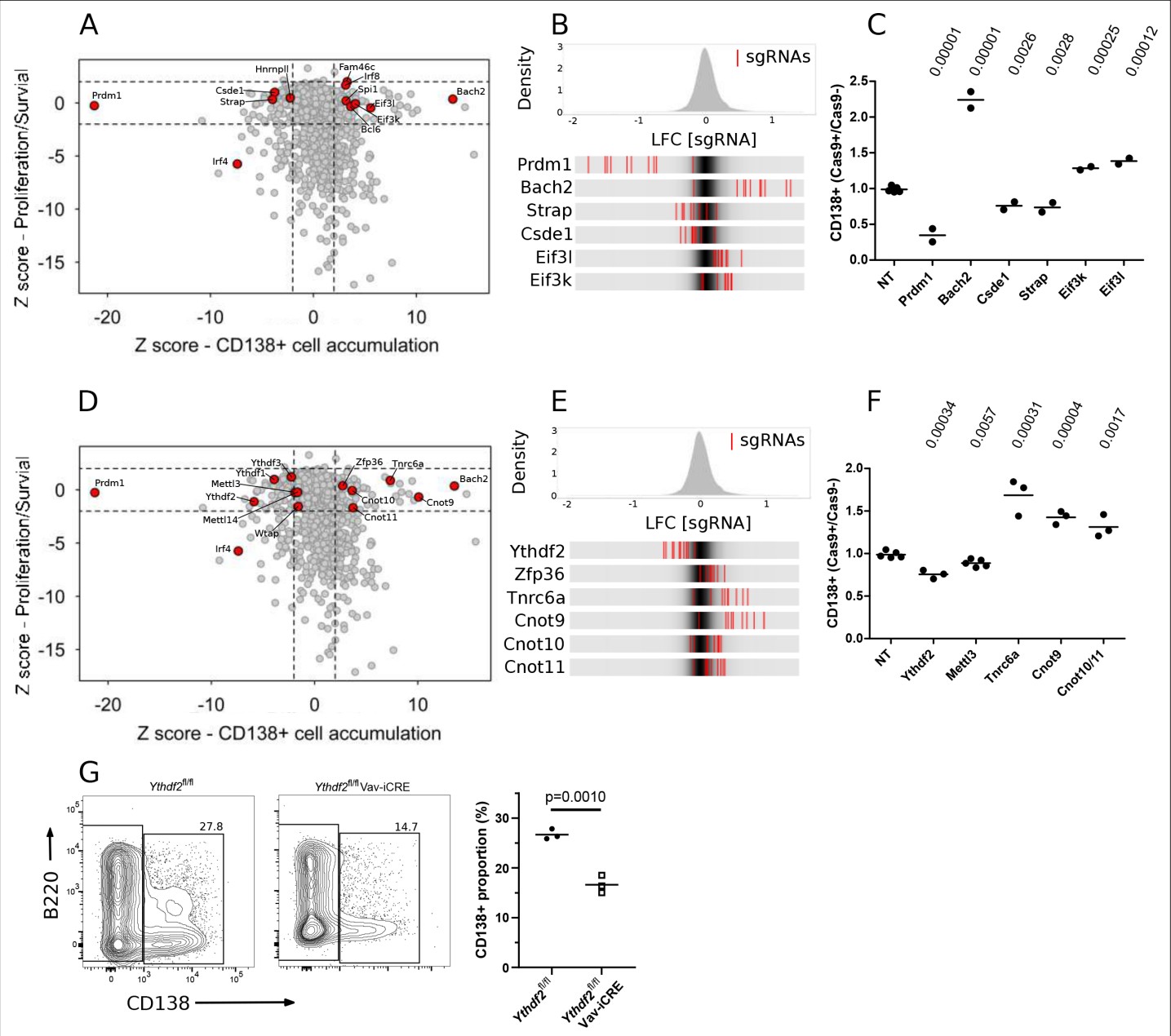

**Figure 1.** Genetic screen identifies modulators of B cell expansion and plasma cell accumulation. (**A**) Dot plot representation of genetic screens of B cell proliferation/survival, and CD138 + cell accumulation: X-axis shows z-score of gene-level log$_2$ fold change (LFC) for CD138 + cell accumulation (CD138 +v CD138- cells) and Y-axis shows z-score of gene-level LFC (day 8 v day 4 cells) for B cell expansion calculated by MAGECK. (**B**) Top: Distribution of enrichments of all sgRNAs (z-scores of sgRNA-level LFC: CD138 +v CD138- cells) for the CD138 + cell accumulation screen. Bottom: Enrichment for each of the ten sgRNAs, represented as red lines, targeting one of the indicated genes in comparison to the overall distribution of sgRNAs in the screen, depicted as the grey density in the middle of each bar. (**C**) The ratio of the proportion of cells expressing CD138 in Cas9 + cells and Cas9- cells transduced by viruses with non-targeting (NT), *Prmd1, Bach2, Csde1, Strap, Eif3k, Eif3l* targeting sgRNAs at day-8 of the in vitro B cell culture the data are representative of between two and four experiments performed on different days. Statistical significance was determined by two-tailed unpaired Student's t-test, p values unadjusted for multiple testing are plotted. Each symbol is representative of a distinct sgRNA. (**D**) Same as in (**A**) with additional genes highlighted. (**E**) Same as in (**B**) with additional genes highlighted. (**F**) Same as in (**C**) with *Ythdf2, Mettl3, Tnrc6a, Cnot9* targeting sgRNAs and paired sgRNAs against *Cnot10/Cnot11*. the data are representative of between two and four experiments performed on different days. (**G**) Left: representative flow cytometry for *Ythdf2* CTL and *Ythdf2* CKO and right: summary data of the proportion of cells expressing CD138 at day 8 of an in vitro culture of B cells from *Ythdf2*$^{fl/fl}$ mice (closed circles) or *Ythdf2*$^{fl/fl}$-*Vav1*-iCre mice (open squares); the data are representative of three experiments performed on different days. Statistical significance was determined by two-tailed unpaired Student's t-test. Each symbol is representative of cells from a single mouse.

*Figure 1 continued on next page*

*Figure 1 continued*

The online version of this article includes the following figure supplement(s) for figure 1:

**Figure supplement 1.** Custom sgRNA library targeting RBPs identifies regulators of B cell proliferation and survival.

accumulation and the eIF3 octamer subunits eIF3K and eIF3L as limiting CD138 + cell accumulation (*Figure 1C*, *Figure 1—figure supplement 1H, I*). Loss-of-function mutations in eIF3K and eIF3L have previously been shown to extend *Caenorhabditis elegans* lifespan by enhancing resistance to endoplasmic reticulum (ER) stress (*Cattie et al., 2016*). Hence, eIF3K and eIF3L may inhibit CD138 + cell appearance by limiting tolerance to ER stress. All three YTHDF-family members of m⁶A binding proteins were also enriched for increasing CD138 + cell abundance (*Figure 1D*) of which *Ythdf2* had the most pronounced effect (*Figure 1E, F*; *Figure 1—figure supplement 1H, I*). In further support of a role of m6A in promoting CD138 + cell accumulation, the three core components of the m6A writer complex (*Mettl3*, *Mettl14* and *Wtap*), had convergent enrichment close to our cutoff for significance as promoting CD138 + cell abundance (*Figure 1D, F*). By contrast, ZFP36, an AU-rich element binding protein, and the TNRC6 family critical for miRNA mediated repression were enriched for limiting CD138 + cell abundance (*Figure 1D–F* and *Figure 1—figure supplement 1H, I*). ZFP36 has been previously reported to limit CD138 + cell appearance in vitro (*Chu et al., 2016*).

The YTHDF-, ZFP36- and TNRC6-families of RBPs are all known to promote RNA decay by recruiting the CCR4-NOT complex leading to deadenylation and subsequent decay (*Du et al., 2016*; *Fabian et al., 2013*; *Chen et al., 2014*). Interestingly, CNOT9, and the interacting partners CNOT-10 and –11 which form a CNOT submodule were enriched for limiting CD138 + cell abundance (*Figure 1D–F*). We hypothesise that the CCR4-NOT complex via its interactions with different adapters may act as a post-transcriptional hub that controls survival or the decision to differentiate. In this model, we envisage the ZFP36- and TNRC6-families suppress transcripts that promote plasma cell differentiation or survival, while the YTHDF-family suppress transcripts that inhibit plasma cell differentiation or survival.

## Antibody secreting cells deficient for YTHDF2 fail to accumulate

Our screen demonstrated that *Ythdf2* and its paralogues promote CD138 + cell accumulation, without appreciably affecting B cell expansion. To confirm this, we took advantage of *Ythdf2*ᶠˡ/ᶠˡ;*Vav1-iCre* (*Ythdf2* CKO) mice, in which *Ythdf2* is conditionally deleted from the haematopoietic system (*Paris et al., 2019*; *Mapperley et al., 2021*). Following anti-IgM stimulation in vitro the numbers of *Ythdf2* CKO B cells at each division was not different from *Ythdf2*ᶠˡ/ᶠˡ control (*Ythdf2* CTL) B cells (*Figure 1—figure supplement 1J*). Moreover, *Ythdf2* CKO B cells showed no difference in the proportions of IgG1 +and IgE + cells generated in vitro compared to *Ythdf2* CTL B cells (*Figure 1—figure supplement 1K*), indicating that class switch recombination (CSR) did not require YTHDF2 in vitro. Because *Ythdf2* CKO B cells generated fewer CD138 + cells compared to *Ythdf2* CTL B cells (*Figure 1G*), we hypothesise that YTHDF2 promotes B cell differentiation independently of a role in CSR, cell proliferation or survival, or specifically promotes the survival of differentiated CD138 + cells.

To investigate an in vivo role of *Ythdf2*, we established mixed bone-marrow chimeras in B6.SJL recipients with μMT and either *Ythdf2* CTL or *Ythdf2* CKO bone marrow cells. In these chimeras, CD45.1+ μMT stem cells, which cannot generate B cells, ensure that non-B cells are primarily *Ythdf2*-sufficient, whereas all B cells derive from CD45.2 + control or experimental donors. *Ythdf2*-deficient cells efficiently reconstituted the splenic B cell pool (*Figure 2A*) enabling us to test the hapten-specific responses of mature B cells lacking YTHDF2 in two independent immunisation experiments with alum precipitated 4-Hydroxy-3-nitrophenylacetyl coupled to the carrier protein Keyhole-Limpet Hemocyanin (NP-KLH). At 21 days after immunisation the ratio of high and low valency NP-binding IgG1 in *Ythdf2* CKO chimeras was comparable to controls (*Figure 2B*), however, a slight defect may exist that remains undetected by the power of our experiment. At day 21, the number of CD138 +CD267+ (*Figure 2—figure supplement 1A*) NIP +IgG1 + splenic plasmablasts (*Figure 2C*) and NIP +IgG1 + plasma cells in the bone marrow (*Figure 2D*) were both threefold lower in the *Ythdf2* CKO compared to the controls. We observed no difference in CXCR4 expression in CD138 +CD267 + cells between genotypes (*Figure 2—figure supplement 1B*). Together these results demonstrate a role for YTHDF2 in the accumulation of plasma cells in vivo.

As an independent experimental approach of addressing the role of YTHDF2 in plasma cell accumulation, we also generated competitive mixed bone marrow chimeras in lethally irradiated B6.SJL

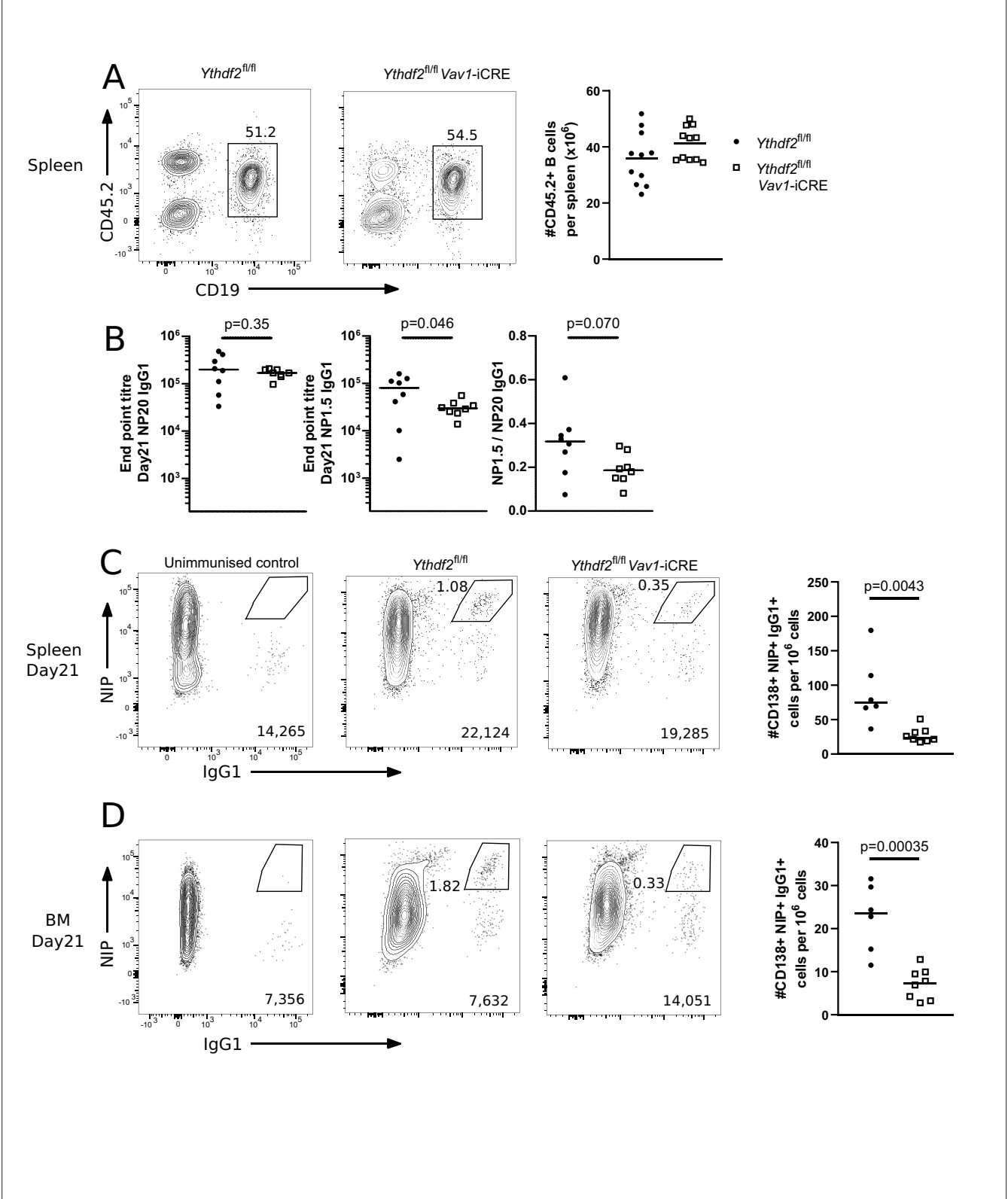

**Figure 2.** Analysis of μMT chimeras at day 21 after immunisation with NP-KLH in alum. (**A**) Left: Representative flow cytometry analysis of cells expressing CD45.2 and CD19 in spleen of μMT chimeras 14 weeks after reconstitution. Numbers refer to the proportion of viable single cells within CD45.2 + CD19 + gate. Right: The number of cells expressing CD45.2 and CD19, n = 11. (**B**) Left: End point titre of serum for anti-NP20 IgG1 antibody. Center: End point titre of serum for anti-NP1.5 IgG1 antibody Right: Ratio of end point titres of serum for anti-NP1.5 and anti-NP20 IgG1 antibodies.

*Figure 2 continued on next page*

*Figure 2 continued*

Representative flow cytometry analysis of NIP and IgG1 intracellular staining of spleen (**C**) or bone marrow (**D**) cells expressing CD138 and CD267. The numbers in the bottom right corner of left-hand panels of C and D show the number of events plotted. The numbers adjacent to the gates indicate the proportion of NIP +IgG1 + cells. The right-hand panels of C and D show the number of cells expressing NIP and IgG1 per million viable cells as calculated from the indicated gates. Each symbol represents an individual *Ythdf2*^fl/fl^ control (closed circles, n = 6) and *Ythdf2*^fl/fl^-*Vav1*-iCre knockout (open squares, n = 8) mouse. Statistical significance was determined by a two-tailed unpaired Student's t-test. The data are pooled from two separate immunisation experiments.

The online version of this article includes the following figure supplement(s) for figure 2:

**Figure supplement 1.** Identification of CD138 +CD267 + splenic cells by flow cytometry.

recipients with equal amounts of CD45.1 + B6.SJL and either CD45.2 + *Ythdf2* CTL or *Ythdf2* CKO bone marrow cells and performed one immunisation experiment with NP-KLH. This system allows YTHDF2-deficient B cells to be in competition with YTHDF2-sufficient B cells at all developmental stages, thus offering a stringent test for functionality of YTHDF2-deficient cells. The *Ythdf2* conditional allele was engineered to express a GFP-YTHDF2 fusion protein in the absence of Cre recombinase, so that GFP expression reliably indicates cells that express YTHDF2 (*Ivanova et al., 2017*). Previously, *Ythdf2* was reported to have a minor role in promoting B cell development (*Zheng et al., 2020*). In our competitive chimeras, YTHDF2-deficient cells efficiently reconstituted the mature B cell pool (*Figure 3A*) with >98% being GFP negative and thus having undergone Cre-mediated deletion (*Figure 3—figure supplement 1A*). At 21 days after immunisation, the number of NIP +IgG1 + splenic plasmablasts was fivefold lower in the *Ythdf2* CKO chimeras compared to the controls (*Figure 3B*). *Ythdf2* CKO plasma cells were depleted in the bone marrow compartment (*Figure 3C*) and those present were predominantly deficient for *Ythdf2* (*Figure 3—figure supplement 1B*). Moreover, *Ythdf2* CKO antigen specific IgG1 +plasma cells in the bone marrow were also fivefold less abundant in the *Ythdf2* CKO chimeras compared to the controls (*Figure 3D* and *Figure 3—figure supplement 1C*). This defect was specific to the mutation of *Ythdf2* as there were similar numbers of CD45.1 B6.SJL-derived plasma cells (*Figure 3—figure supplement 1D*) and NIP +IgG1 + plasma cells (*Figure 3—figure supplement 1E*) in the bone marrow of *Ythdf2* CTL and *Ythdf2* CKO chimeras. These data are consistent with the findings from the µMT chimeras and suggests *Ythdf2* is required for the accumulation of plasma cells in vivo.

## The spectrum of m⁶A modified transcripts in B cells

To further understand how the m⁶A modification of mRNA may impact B cell activation we performed m⁶A-eCLIP to identify methylated transcripts within the B cell transcriptome. In total 3370 high confidence m⁶A sites were identified among 2658 transcripts in wild-type B cells cultured on 40LB cells for 8 days. These m⁶A sites were strongly enriched for DRACH (D = G/A/U, *R* = G/A, H = A/U/C) and more specifically the RRACU motifs (*Figure 4A*). m6A sites were enriched in the terminal exon and in the location of the STOP codon in accordance with published datasets (*Figure 4B*; *Dominissini et al., 2012*; *Meyer et al., 2012*; *Gerald et al., 2015*). We hypothesised that m⁶A may mark B cell specific transcripts for turnover upon differentiation. To investigate this, we performed RNA-seq of wild-type B cells cultured on 40LB cells in vitro for eight days and sorted by flow cytometry based on B220hi CD138- or B220lo CD138 +expression. However, the m⁶A sites present in the B cell transcriptome (at the 3'UTR and/or CDS) were not characteristic of transcripts specific to either differentiation state and were not biased to transcripts of high expression levels (*Figure 4C*). As YTHDF paralogues are expected to bind all available m⁶A sites in a transcriptome we speculate that YTHDF2 binds transcripts specific to both differentiation states (*Zaccara and Jaffrey, 2020*). However, we cannot rule out the possibility that YTHDF2 has an m⁶A independent role. Among the methylated transcripts identified were *Bach2*, *Pax5*, *Irf8*, and *Spi1* all of which encode transcription factors that inhibit B cell terminal differentiation, and *Prdm1* which promotes differentiation (*Figure 4D*).

In summary, we have provided evidence that hundreds of RBPs regulate the accumulation of CD138 + cells. ZFP36- and TNRC6-families, as well as CNOT9 and the interacting partners CNOT10 and CNOT11 of the CCR4-NOT complex have roles in limiting the accumulation of CD138 + cells. By contrast, the CCR4-NOT binding partner YTHDF2 has an essential role in promoting the accumulation of plasma cells in the spleen and bone marrow. The mechanisms by which these RBPs limit CD138 + cell accumulation remain to be investigated. Our in vitro data support a role for YTHDF2 in promoting

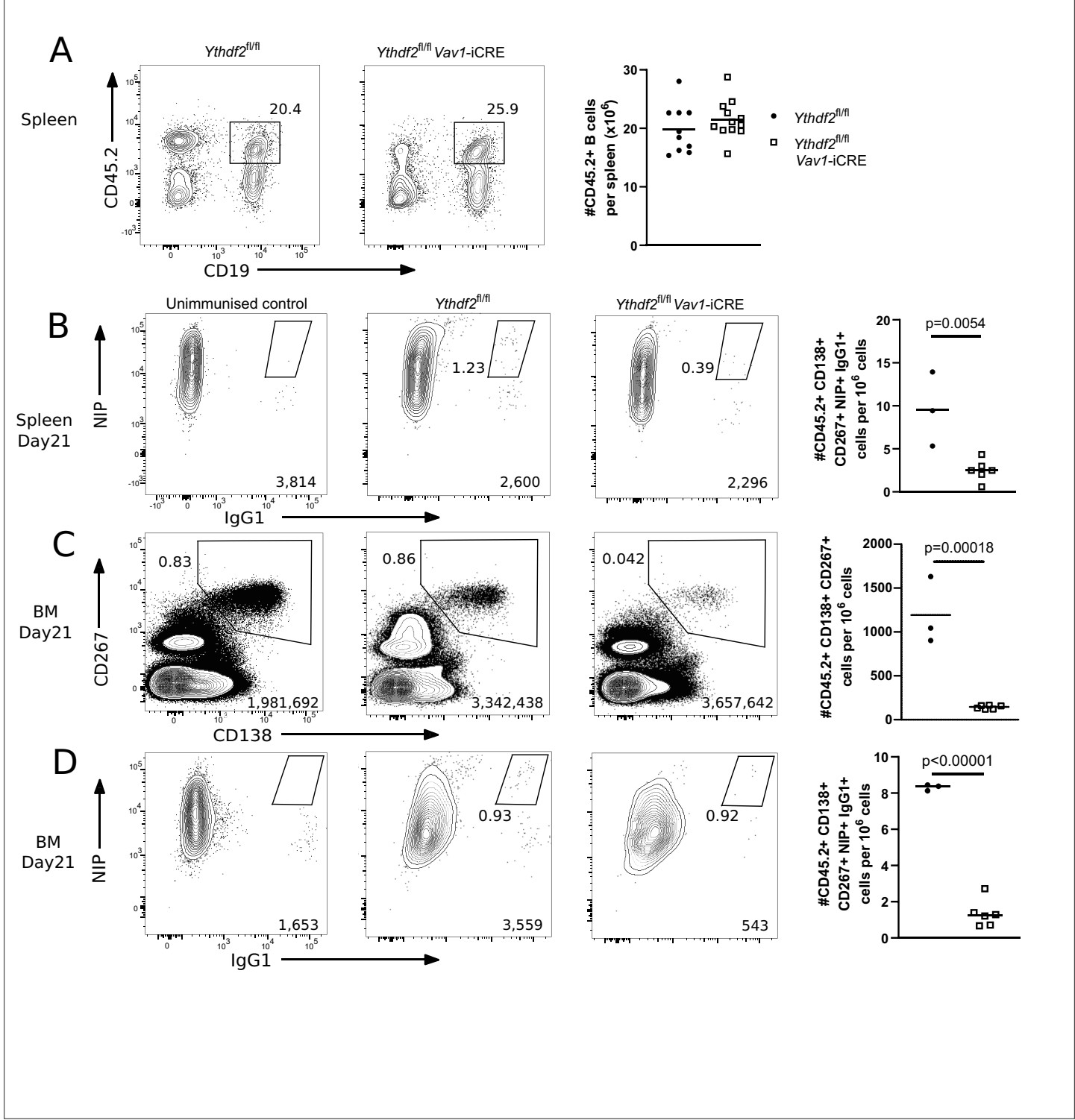

**Figure 3.** YTHDF2 deficient plasma cells fail to accumulate in the bone marrow. (**A**) Left: Representative flow cytometry analysis of cells expressing CD45.2 and CD19 in spleen of competitive chimeras twelve weeks after reconstitution. Numbers refer to the proportion of viable single cells within CD45.2 + CD19 + gate. Right: The number of cells expressing CD45.2 and CD19, n ≥ 10. (**B**) Representative flow cytometry analysis of NIP and IgG1 intracellular staining on spleen cells gated as CD45.2 + CD138+, CD267 + cells at day 21. (**C**) Representative flow cytometry analysis of CD138 and CD267 staining on bone marrow cells gated as CD45.2 + at day 21. (**D**) Representative flow cytometry analysis of NIP and IgG1 intracellular staining on bone marrow cells gated as CD45.2 + CD138+, CD267 +at day 21. For each condition, the numbers in the bottom right corner of left-hand panels show the number of events plotted. The numbers adjacent to the gates indicate the proportion of cells within the gate. For each condition, the right-hand

*Figure 3 continued on next page*

*Figure 3 continued*

plots show the number of cells per million viable cells. Symbols represent data from an individual *Ythdf2*^fl/fl^ control (closed circles, n = 3) and *Ythdf2*^fl/fl^-*Vav1*-iCre knockout (open squares, n = 6) mouse. Statistical significance was determined by two-tailed unpaired Student's t-test. The data are from a single immunisation experiment.

The online version of this article includes the following figure supplement(s) for figure 3:

**Figure supplement 1.** Absence of cells that escaped Cre mediated deletion.

plasma cell accumulation via differentiation or survival, while our in vivo bone marrow data could also be explained by a role for YTHDF2 in plasma cell migration. We also observed a defect in *Ythdf2* deficient plasma cell accumulation in the spleen and identified m6A methylation on transcripts encoding transcriptional repressors and promotors of plasma cell differentiation; together these indicate that YTHDF2 mediated regulation of differentiation or migration are not sufficient to fully explain YTHDF2 support of plasma cell accumulation. One tempting hypothesis is that YTHDF2 promotes plasma cell accumulation by limiting apoptosis induced by ER stress (*Einstein et al., 2021*) specifically within differentiated plasma cells. Furthermore, METTL3 is required for GC formation and YTHDF2 has been previously shown to have a role in supporting the germinal centre reaction (*Grenov et al., 2021*), which may promote plasma cell accumulation. Determining the precise mechanism by which YTHDF2 impacts plasma cell biology requires further studies of differentiation, survival and migration.

# Materials and methods
## Construction of sgRNA vector backbone and mouse RBP sgRNA library
MSCV_hU6_BbsI-ccdB-BbsI_iScaffold_mPGK_puro-2A-CD90.1 was generated in one round of cloning from a similar plasmid previously generated in our lab (*Turner and Turner, 2021*) (iScaffold refers to an improved sgRNA scaffold design *Dang et al., 2015*). A PCR product encoding hU6_BbsI-ccdB-BbsI_iScaffold_mPGK_puro-2A-CD90.1 and the appropriate flanking sequences was generated and ligated by Gibson assembly into a XhoI +SalI linearised vector (MIGRI). Our mouse RBP sgRNA library was generated as previously described (*Turner and Turner, 2021*). For individual sgRNAs, two 24nt oligonucleotides were annealed and ligated by T4 ligation into our BbsI linearised MSCV backbone vector. For pairwise sgRNAs, PCR products encoding hU6_sgRNA1 and mU6_sgRNA2 with the appropriate flanking sequences were generated and ligated by Gibson assembly into a BbsI linearised RetroQ_BbsI-ccdB-BbsI_iScaffold_mPGK_CD90.1 vector. We targeted mouse RBPs identified by multiple high-throughput studies employing mRNA interactome capture (RIC) (*Castello et al., 2012*; *Baltz et al., 2012*; *Beckmann et al., 2015*; *Gerstberger et al., 2014*; *Kwon et al., 2013*; *He et al., 2016*; *Liepelt et al., 2016*). Our library targets RBPs identified in at least two mouse RIC studies; the mouse orthologues of RBPs identified in at least two human RIC studies; and manually curated RBPs identified in only one RIC study that had "RNA binding" gene ontology or were paralogues of a human RBP orthologue.

## In vitro plasma cell differentiation
3T3 cells expressing mouse CD40L and human BAFF (40LB cells *Nojima et al., 2011*) were maintained in Roswell Park Memorial Institute 1640 (RPMI-1640) media supplemented with 10% FBS and 1 x GlutaMAX (Gibco: 35050061). Irradiated (~120Gγ) 40LB cells were seeded at $80 \times 10^3$ cells per well in a 12-well plate in supplemented RPMI-1640 media. The next day, naive B cells were isolated from the spleens of mice using the B cell isolation kit (Miltenyi Biotec: 130-090-862) and $30 \times 10^3$ cultured on the irradiated 40LB cells with IL-4 in supplemented RPMI-1640 media (final: 10% FBS, 1 x GlutaMAX, 50 μM 2-mercaptoethanol Thermo Scientific: 31350–010, 100 units/ml penicillin, and 100 μg/ml streptomycin Thermo Scientific: 15140, 10 ng/ml mIL-4 PeproTech 214–14). On day 4 B cells were reseeded on freshly irradiated 40LB cells with IL-21 in supplemented RPMI-1640 media (final: 10% FBS, 1 x GlutaMAX, 50 μM 2-mercaptoethanol, 100 units/ml penicillin, and 100 μg/ml streptomycin, 10 ng/ml mIL-21 PeproTech 210–21). On day 8, B cells were harvested and analysed. 40LB cells were a kind gift from Prof D. Kitamura, they were negative for mycoplasma contamination, and authenticated for CD40L expression by flow cytometry.

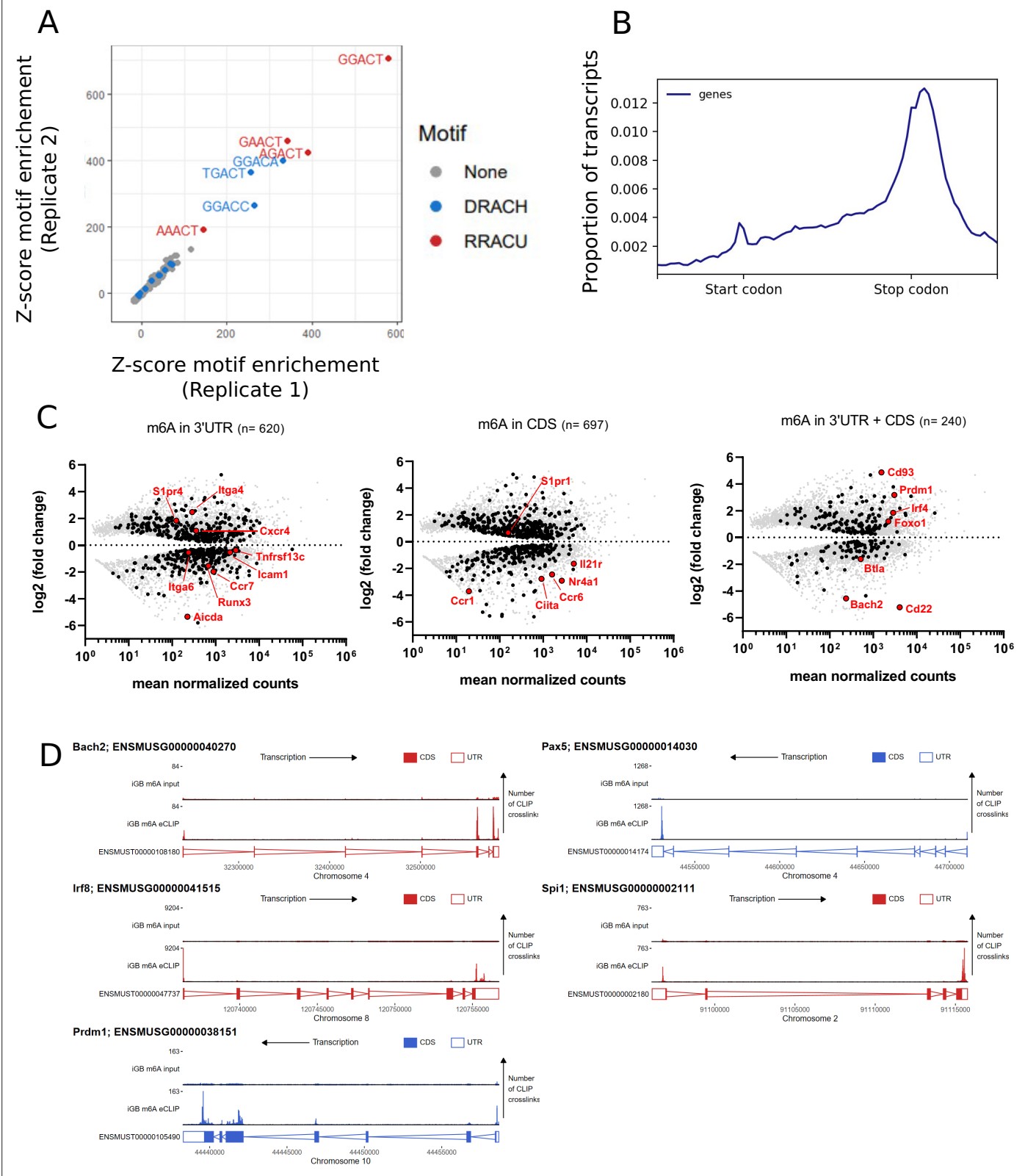

**Figure 4.** Negative regulators of B cell differentiation are methylated. (**A**) Z-scores showing enrichment of five base motifs centred at in vitro derived B cell m6A-eCLIP crosslink sites relative to randomised control sites in each replicate, n = 2. (**B**) Metagene analysis of the location of m6A clusters throughout the B cell transcriptome. (**C**) Log2 fold change for differentially expressed (FDR-adjusted p-values < 0.05) methylated (black - identified using the CLIPper pipeline) and unmethylated (grey) transcripts between CD138 +B220 lo and CD138- B220hi in vitro cultured cells, n = 3. (Left) 3′UTR

*Figure 4 continued on next page*

*Figure 4 continued*

methylated transcripts, (centre) CDS methylated transcripts, (right) transcripts methylated in both their 3′UTR and CDS. (**D**) Methylation pattern of individual transcripts encoding regulators (*Bach2, Pax5, Irf8, Spi1, Prdm1*) of B cell differentiation.

## Retrovirus production and transduction

The Plat-E retrovirus packaging cell line was maintained in supplemented Dulbecco's modified Eagle's (DMEM) high glucose media (Gibco: 41965) with 10% fetal bovine serum (FBS). Plat-E cells in logarithmic growth phase were seeded in dishes (Nunc 150350). The next day, a transfection mix of 1000 µl OptiMEM (Gibco: 31985), 30 µl TransIT–293 (Mirus Bio: MIR2700), 9 µg transfer vector, 2 µg packaging pCL-ECO vector was added dropwise and incubated with the cells overnight. The media was replaced. Then, for 2 sequential days the viral supernatant was harvested. B cells cultured on 40LBs for 3 days were transduced with retrovirus particles encoding sgRNAs in the presence of 4 ng/ml polybrene (Sigma-Aldrich: H9268), by spin-fection at 1000 g for 45 min at 32 °C.

## CRISPR/Cas9 genetic screen of B cell differentiation, proliferation, and survival

B cells were cultured with 40LB, on day 3 cells were transduced with the mouse RBP sgRNA library at an MOI of 0.1 with a predetermined amount of retrovirus supernatant. On day 4, transduced cells were positively selected by magnetic assisted cell sorting (MACS) via the cell surface antigen CD90.1 (Thy1.1) using microbeads (Miltenyi Biotec: 130-094-523); the manufacturer's protocol was adjusted by using 4 x less reagents and by maintaining cells at room temperature. On day 8, CD138- and CD138 +B cells were physically separated by fluorescence activated cell sorting, washed in PBS, and snap frozen on dry ice in a 15 ml conical tube before storage at –80 °C for future analysis of CD138 + cell accumulation. In addition, on day 4 and day 8, bulk B cells, unselected for transduction or CD138 expression, were processed and stored in a comparable fashion for future analysis of B cell proliferation/survival. Enough cells to ensure a representation of >1000 x transduced cells per sgRNA in the library were maintained at all timepoints and in all populations.

## Next generation sequencing library generation for CRISPR screen

Genomic DNA (gDNA) was isolated as previously described (*Chen et al., 2015*). Next generation sequencing (NGS) libraries were generated as previously described (*Joung et al., 2017*). Multiplexed NGS libraries were sequenced with an Illumina HiSeq with a 100 bp single end read. The start of the iCRISPR scaffold sequence (GTTTAAGAGCTAT) within each read was identified, and the reads trimmed to encompass the 19 bases immediately preceding this sequence. Bowtie (*Langmead et al., 2009*) was used to map these sequences with zero mismatches to a custom genome comprising the sgRNA sequences, and Seqmonk used to quantify the abundance of each sgRNA (https://www.bioinformatics.babraham.ac.uk/projects/seqmonk/). Analysis of our genetic screens was performed with the MAGeCK software (*Li et al., 2014*). MAGeCK determined gene-level LFC for the proliferation/survival (day 8 cells v day 4 cells) and plasma cell abundance (CD138 + cells v CD138- cells) screens. The non-targeting negative control sgRNAs were used to build a mean-variance model for null distribution, which in turn was used to calculate significant sgRNA enrichment. Positive and negative enrichments were calculated, then independently used in robust rank aggregation to obtain gene-level scores. This process identified the extent to which each sgRNA enriched out of the distribution of negative control sgRNAs, then ranked genes by the consistency of their sgRNAs to outperform the null hypothesis (that they would not enrich outside the NT-sgRNAs). Gene-level p-values were calculated by randomizing sgRNA to target gene allocation in permutation tests; false discovery was controlled by the Benjamini–Hochberg procedure. Log2 fold changes (LFC) were calculated as the median LFC for all sgRNAs for a targeted gene that enriched in the same direction. Z-scores were calculated as the number of standard deviations from the mean of all negative control LFCs. Our statistical cutoff for gene level enrichment was set at a Z-score of greater than or equal to +/-2, and an FDR adjusted p-value of less than 0.05.

## Validation of targets from CRISPR/Cas9 knockout screens

A mixed population of GFP-Cas9 positive and GFP-Cas9 negative B cells, at a ratio of 75:25, were cocultured and transduced with individual sgRNA vectors against genes of interest. The ratio of transduced cells that were differentiated between the Cas9 positive and Cas9 negative populations was calculated and used to control for the variation of the in vitro germinal centre B cell culture between experiments. Furthermore, the change in GFP+: GFP- ratio over the culture indicated whether genetically modified cells (Cas9+) had a competitive advantage over unperturbed cells (Cas9-).

## Proliferation assay

Na-ve B cell were labelled with CellTrace Violet (Thermo Scientific: C34557), following the manufacturer's protocol, then seeded at $2 \times 10^6$ cells/ml in 100 µl per well of a 96-well plate and stimulated with anti-IgM (2.5 µg/ml) (F(ab')$_2$ fragment, polyclonal) (Jackson ImmunoResearch: 115-006-020) with IL-4 (10 ng/ml) and IL-21 (10 ng/ml) (PeproTech: 210–21). The inclusion of counting beads enabled the calculation of absolute cell numbers.

## Mice

All mice were on a C57BL/6 background. R26-GFP-Cas9 Mb1-Cre mice were derived from C57BL/6$^{Cd79atm1(cre)Reth}$ (*Hobeika et al., 2006*) and C57BL/6$^{Gt(ROSA)26Sortm1(CAG-cas9*,-EGFP)Fezh}$ (*Platt et al., 2014*). For µMT bone marrow chimera experiments B6.SJL-*Ptprc$^a$Pepc$^b$*/$^{Boy}$ (B6.SJL) mice were used as recipients and reconstituted with CD45.1 µMT (Ighm$^{tm1Cgn}$) (*Kitamura et al., 1991*) and either *CD45.2 Ythdf2* CTL (*Ythdf2$^{Tg(Vav1-icre)A2Kio}$*) or *CD45.2 Ythdf2* CKO (*Ythdf2$^{tm1.1Doca + Tg(Vav1-icre)A2Kio}$*) mice (*Paris et al., 2019*; *Mapperley et al., 2021*; *Ivanova et al., 2017*). For competitive bone marrow chimeras, B6.SJL-*Ptprc$^a$Pepc$^b$*/$^{Boy}$ (B6.SJL) mice were used as recipients and reconstituted with B6.SJL mice and either *CD45.2 Ythdf2* CTL or *CD45.2 Ythdf2* CKO mice.

Ethics Statement: All mouse experimentation was approved by the Babraham Institute Animal Welfare and Ethical Review Body and was licensed by the United Kingdom Home Office under PPL P4D4AF812. Mice were bred and maintained in the Babraham Institute Biological Support Unit. Since the opening of this barrier facility in 2009 no primary pathogens or additional agents listed in the FELASA recommendations have been confirmed during health monitoring surveys of the stock holding rooms. Ambient temperature was ~19–21 °C and relative humidity 52%. Lighting was provided on a 12 hr light: 12 hr dark cycle including 15 min 'dawn' and 'dusk' periods of subdued lighting. After weaning, mice were transferred to individually ventilated cages with 1–5 mice per cage. Mice were fed CRM (P) VP diet (Special Diet Services) ad libitum and received seeds (e.g. sunflower, millet) at the time of cage-cleaning as part of their environmental enrichment.

## Generation and primary immunisation of chimeric mouse models

B6.SJL recipient mice were lethally irradiated with two doses of 5.0 Gy. For µMT chimeras, recipients were reconstituted with $3 \times 10^6$ total bone marrow cells composed from 80% µMT donor bone marrow cells and 20% either *Ythdf2* CTL or *Ythdf2* CKO donor bone marrow cells. For competitive chimeras, recipients were reconstituted with $3 \times 10^6$ total bone marrow cells composed from 50% B6SJL donor bone marrow cells and 50% either *Ythdf2* CTL or *Ythdf2* CKO donor bone marrow cells. Mice received intraperitoneal injection of 200 µl sterile PBS containing 100 µg of NP(23)KLH adsorbed in 40% v/v Alum (Serva: 12261). NP-KLH was emulsified in Alum by rotation for 30 min at room temperature protected from light.

## ELISA

NP-specific antibodies were detected by ELISA as previously described (*Saveliev et al., 2021*); antibody end point titres were used as a measure of relative concentration.

## Flow cytometry

Cell suspensions from spleen and bone marrow were prepared and stained as previously described (*Saveliev et al., 2021*). NIP and IgG1 expression in CD138 +CD267 + plasma cells was detected with intracellular staining. A list of antibodies is provided in *Supplementary file 1F*.

## m⁶A-eCLIP

B cells were cultured in vitro and on day 8 total B cell and plasma cells were separated from 40LB cells by negative selection with H-2Kd-biotin antibody and anti-biotin microbeads (Miltenyi 130-090-485) following the manufacturer's instructions. Total RNA was prepared with a kit (Zymo R2050) following the manufacturer's instructions and sent to Eclipse Bioinnovations who isolated poly(A) RNA, cross-linked an m⁶A antibody, performed immunoprecipitation and NGS.

Data were analysed by Eclipse Bioinnovations according to their standard m⁶A-eCLIP analysis pipeline (https://eclipsebio.com/wp-content/uploads/2021/06/eclipsebio_data_analysis_review_m⁶A-eCLIP.pdf). Briefly: The first 10 bases of each read (which comprise a UMI) were trimmed using UMI-tools (*Smith et al., 2017*; *Li et al., 2011*), followed by trimming of poor quality and adaptor sequences from the 3' end of the reads using cutadapt. Reads mapping to repetitive elements were filtered out, and the remaining reads mapped to the mouse GRCm38 genome build using STAR. PCR duplicates were removed using UMI-tools. Clusters of reads (peaks of m⁶A modification) were iden-tified using CLIPper (https://github.com/YeoLab/clipper/wiki/CLIPper-Home), and log2 fold change relative to the corresponding input sample calculated. Clusters that were reproducible between repli-cates were then identified using IDR (*Krakau et al., 2017*) (https://github.com/nboley/idr). Single nucleotide resolution crosslink sites with enrichment relative to input were also identified using Pure-CLIP (*Ramírez et al., 2016*). The gene or feature type to which each crosslink site was assigned was based on the Ensembl Mouse GRCm38.97 annotation release. If the site overlapped multiple genes or feature types, the most likely was chosen in a hierarchical way: first, protein coding isoforms were prioritised; second, transcript isoforms with support level <4 were prioritised; third by a hierarchy of feature types (CDS >3'UTR >5'UTR >intron > non-coding exon >non-coding intron); third, higher confidence isoforms (based first on transcript support level and second on whether they have a CCDS) were prioritised.

To assess enrichment of 5 base motifs at the crosslink sites, the 5 bases centred on each cross-link site for each replicate were identified, and the representation of all 5-mers quantified. This was repeated for 100 sets of control sites, where the locations of the crosslink sites were randomised across all genes that contain at least one crosslink site, ensuring the same distribution across features as in the m⁶A-eCLIP dataset. The z-score for each 5mer was then calculated as: (occurrence at eCLIP crosslink sites - mean occurrence at control sites) / standard deviation of occurrence at control sites.

Metagene analysis of the distribution of m⁶A was performed on the Galaxy server using deepTools (*Kim et al., 2015*). The computeCoverage tool was first used using the Ensembl Mouse GRCm38.97 GTF file, scaling all CDS annotations and additionally analysing 1 kb up- and downstream of each. A BED file containing the reproducible clusters was used as the score file, and the maximum value calculated, without skipping 0 s (therefore, this will be 1 for windows containing a cluster and 0 for those without). The resulting matrix was then used as input to plotProfile, and the average across all genes plotted.

## Generation and analysis of mRNA sequencing libraries

For transcriptomic analysis, total RNA was isolated from 0.4 × 10⁶ FACS-sorted B220hi CD138- and B220lo CD138 +B cells (derived from 3 biological replicates after eight days in culture) using the RNeasy Mini Kit (Qiagen). cDNA was generated from polyadenylated transcripts employing the SMART-Seq v4 ultra low input RNA kit (Takara Bio). RNA and cDNA quality was analysed on a 2100 Bioanalyser (Agilent). mRNAseq libraries were prepared using Nextera XT DNA library preparation kit (Illumina) and quantified with KAPA library quantification kit (Roche). Barcoded libraries were multi-plexed and sequenced on an Illumina HiSeq 2500-RapidRun system on a 50 bp single-end mode with a coverage of 20 M reads per sample. Reads were trimmed using Trim Galore and mapped to mouse genome GRCm38.97 using HiSat2 (2.1.0) (*Love et al., 2014*). Raw counts were calculated over mRNA features (excluding Ig molecules) using SeqMonk (1.47.2; https://www.bioinformatics.babraham.ac.uk/projects/seqmonk/). DESeq2 (1.30.1)[60] was used to calculate differential RNA abun-dance and performed using default parameters, with 'normal' log2 fold change shrinkage. p Values were adjusted for false discovery rate using the Benjamini-Hochberg method. Information on biolog-ical replicates were included in the design formula to have paired analysis. Differentially expressed transcripts were cross-referenced with transcripts that contained peaks of m⁶A modification identified using CLIPper.

## Acknowledgements

We thank the Babraham Institute Biological Support Unit, Flow Cytometry and Bioinformatics Facilities for outstanding support, and D Hodson and C Ribeiro de Almeida for helpful discussions and comments on the manuscript. This study was supported by funding from the Biotechnology and Biological Sciences Research Council (BBSRC) (BBS/E/B/000C0427; BBS/E/B/000C0428); the BBSRC Core Capability Grant to the Babraham Institute; a BBSRC-iCASE studentship BB/L016745/1 in partnership with Abzena; and a Wellcome Investigator award (200823/Z/16/Z) to MT FS was supported by European Molecular Biology Organization (EMBO) Long-Term Fellowship (ALTF 880–2018). KR K's laboratory is funded by a Cancer Research UK program grant (C29967/A26787) and project grants from the Medical Research Council, Blood Cancer UK, Barts Charity, and the Kay Kendall Leukaemia Fund. We thank the Finkelstein Lab (https://github.com/finkelsteinlab) for the BioRxiv template.

## Additional information

### Funding

| Funder | Grant reference number | Author |
|---|---|---|
| Biotechnology and Biological Sciences Research Council | BBS/E/B/000C0427 | Martin Turner |
| Biotechnology and Biological Sciences Research Council | BBS/E/B/000C0428 | Martin Turner |
| Wellcome Trust | 200823/Z/16/Z | Martin Turner |
| Biotechnology and Biological Sciences Research Council | BB/L016745/1 | David J Turner |
| European Molecular Biology Organization | ALTF 880-2018 | Fiamma Salerno |
| Cancer Research UK | C29967/A26787 | Kamil R Kranc |
| European Molecular Biology Organization | | Kamil R Kranc |
| Babraham Institute | | Kamil R Kranc |
| Kay Kendall Leukaemia Fund | | Kamil R Kranc |

The funders had no role in study design, data collection and interpretation, or the decision to submit the work for publication. For the purpose of Open Access, the authors have applied a CC BY public copyright license to any Author Accepted Manuscript version arising from this submission.

### Author contributions

David J Turner, Conceptualization, Data curation, Formal analysis, Investigation, Methodology, Project administration, Visualization, Writing – original draft, Writing – review and editing; Alexander Saveliev, Conceptualization, Investigation, Methodology, Writing – review and editing; Fiamma Salerno, Conceptualization, Formal analysis, Visualization, Writing – review and editing; Louise S Matheson, Conceptualization, Data curation, Formal analysis, Visualization, Writing – review and editing; Michael Screen, Conceptualization, Methodology, Validation, Writing – review and editing; Hannah Lawson, David Wotherspoon, Resources; Kamil R Kranc, Conceptualization, Funding acquisition, Supervision, Writing – review and editing; Martin Turner, Conceptualization, Funding acquisition, Project administration, Supervision, Writing – review and editing

### Author ORCIDs

David J Turner http://orcid.org/0000-0003-4379-6152
Martin Turner http://orcid.org/0000-0002-3801-9896

### Ethics

All mouse experimentation was approved by the Babraham Institute Animal Welfare and Ethical Review Body and was licensed by the United Kingdom Home Office under PPL P4D4AF812.

### Decision letter and Author response

Decision letter https://doi.org/10.7554/eLife.72313.sa1
Author response https://doi.org/10.7554/eLife.72313.sa2

---

## Additional files

### Supplementary files

- Transparent reporting form
- Supplementary file 1. Supplementary tables.

### Data availability

The sgRNA library is available upon request and from Addgene (#169082). The CRISPR/Cas9 knockout screen data and m6A-eCLIP data that support the findings of this study have been deposited in GEO with the GSE179919 accession code, and the RNA-seq data has been deposited in GEO with the GSE179281 accession code.

The following datasets were generated:

| Author(s) | Year | Dataset title | Dataset URL | Database and Identifier |
|---|---|---|---|---|
| Turner DJ, Matheson LS, Turner M | 2021 | The RNA m6A binding protein YTHDF2 promotes the B cell to plasma cell transition | https://www.ncbi.nlm.nih.gov/geo/query/acc.cgi?acc=GSE179919 | NCBI Gene Expression Omnibus, GSE179919 |
| Salerno F, Matheson LS, Turner M | 2021 | Transcriptomic analysis of in vitro induced germinal centre-like B cells and plasmablast differentiation | https://www.ncbi.nlm.nih.gov/geo/query/acc.cgi?acc=GSE179281 | NCBI Gene Expression Omnibus, GSE179281 |

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
