## [Editor Report]

This paper utilizes an elegant Crispr-Cas9 screen to identify RNA binding proteins that may regulate B cell differentiation. With some additional work to verify that the identified proteins are important in vivo, the paper will be of interest to a broad audience of immunologists studying the signals regulating B cell differentiation during an immune response.

---

## [Decision Letter]

**Decision letter after peer review:**

Thank you for submitting your article "The RNA m6A binding protein YTHDF2 promotes the B cell to plasma cell transition." for consideration by *eLife*. Your article has been reviewed by 3 peer reviewers, and the evaluation has been overseen by a Reviewing Editor and Satyajit Rath as the Senior Editor. The following individual involved in review of your submission has agreed to reveal their identity: Koji Tokoyoda (Reviewer #1).

Essential revisions:

1. The discrepancy of Figure 2C and 3D is confusing. Does YTHDR2 affect the formation of germinal centre B cells? If the authors just claim that GC can be formed, to support the cytometric data, histological analyses should be performed.

2. Was the detection of NIP/IgG1 in CD138+ plasma cells performed by intracellular staining? Since most NP-specific IgG1-secreting plasma cells lose surface Igs, the detection may be underestimated. Alternatively, an ELISPOT assay can be also utilized.

3. The identification of YTHDF2-related genes in B/plasma cells should be further assessed. The protein expression of CXCR4, S1PR, integrin alpha4, etc. by flow cytometry will be supportive to Figure 4, even if it makes no difference.

4. Data presented in Figure 1C, 1F, 1G, Figure 2 and Figure 3 should be confirmed in at least 2 independent experiments performed on independent days with appropriate group sizes. The data presented here appears to be from one experimental cohort.

5. For the YTHDF2 knockout experiments did the authors also measure antibody titers to the high valency NP molecules? The data showing a small but significant drop in antibody titers to low-valency NP and if high-valency titers are normal this would suggests affinity maturation maybe affected in the absence of YTHDF2. This experiment would add support to the authors conclusion that the germinal center is not affected in the absence of YTHDF2. Additionally, ELISpot assays should be used as an additional method for enumerating NP-specific plasma cells in the spleens and bone marrow since the numbers that are counted by flow cytometry are so rare.

6.The statistical cutoffs and other metrics (i.e., z-score or fold change) used to define sgRNA enrichment should be clearly stated in the methods section, manuscript text, and/or figure legends.

7. Details and steps for how the CRISPR/Cas9 screen, RNA-seq analysis, and eCLIP data was processed were included in the reporting summary but should also be included in the appropriate section of the manuscript.

8. CD138 expression alone is not necessarily sufficient to identify plasma cells. To determine the relationship between CD138+ cells induced in their in vitro system and in vivo plasma cells, the authors should perform a principal component analysis comparing the gene expression profile obtained from their RNA-seq analysis of CD138+B220lo cells with publicly available gene expression data from plasma cells. The authors could also show the percentage of CD138+B220lo cells that express key markers of plasma cell differentiation such as Blimp1.

9. The authors used mixed bone marrow chimera approaches to determine that YTHDF2 acts in a B cell intrinsic manner. However, these results do not indicate the stage of B cell differentiation where YTHDF2 acts to regulate plasma cell differentiation. Crossing the Ythdf2f/f mice to mice expressing Cre recombinase specifically in germinal center or plasma cells, such as the Jchain Cre, could clarify this issue. Alternatively, the authors could generate chimeras where they transduce Ythdf2f/f bone marrow with a tamoxifen inducible Cre to allow them to specifically ablate Ythdf2 after B cell activation has already occurred.

10. It is also unclear whether YTHDF2 is regulating migration to the bone marrow or maintenance within the bone marrow. A kinetic assessment of the frequency of bone marrow plasma cells in cKO mice could clarify this question.

11. The authors should include the number of times each experiment was repeated, and the total number of mice used in the figure legend. Based on the reporting form it seems that some experiments were only done once with 3-4 mice in the control group. Key experiments should be repeated at least twice with 3-6 mice per group to verify the reproducibility of their results.

*Reviewer #1 (Recommendations for the authors):*

The authors show an important role of an RNA-binding protein (RBP), YTHDF2 in the accumulation of plasma cells. In addition, by a CRISPR/Cas9 knockout screening of RBPs, the authors suggest that some RBPs are involved in plasma cell differentiation. The roles of RBPs in a lymphocyte differentiation system are very interesting, but some technical problems should be solved before publication.

1. The discrepancy of Figure 2C and 3D is confusing. Does YTHDR2 affect the formation of germinal centre B cells? If the authors just claim that GC can be formed, to support the cytometric data, histological analyses should be performed.

2. Was the detection of NIP/IgG1 in CD138+ plasma cells performed by intracellular staining? Since most NP-specific IgG1-secreting plasma cells lose surface Igs, the detection may be underestimated. Alternatively, an ELISPOT assay can be also utilized.

3. The identification of YTHDF2-related genes in B/plasma cells should be further assessed. The protein expression of CXCR4, S1PR, integrin alpha4, etc. by flow cytometry will be supportive to Figure 4, even if it makes no difference.

4. Grenov et al., (bioRxiv, 2020) show that B cell-specific METTL3-deficient mice fail to generate GC B cells and plasma cells. As METTL3 is closed to YTHDF2, the similarities and differences in the two papers should be discussed.

*Reviewer #2 (Recommendations for the authors):*

Data presented in Figure 1C, 1F, 1G, Figure 2 and Figure 3 should be confirmed in at least 2 independent experiments performed on independent days with appropriate group sizes. The data presented here appears to be from one experimental cohort.

For the YTHDF2 knockout experiments did the authors also measure antibody titers to the high valency NP molecules? The data showing a small but significant drop in antibody titers to low-valency NP and if high-valency titers are normal this would suggests affinity maturation maybe affected in the absence of YTHDF2. This experiment would add support to the authors conclusion that the germinal center is not affected in the absence of YTHDF2. Additionally, ELISpot assays should be used as an additional method for enumerating NP-specific plasma cells in the spleens and bone marrow since the numbers that are counted by flow cytometry are so rare.

The statistical cutoffs and other metrics (i.e., z-score or fold change) used to define sgRNA enrichment should be clearly stated in the methods section, manuscript text, and/or figure legends.

Details and steps for how the CRISPR/Cas9 screen, RNA-seq analysis, and eCLIP data was processed were included in the reporting summary but should also be included in the appropriate section of the manuscript.

Multiple typos were found throughout the manuscript that should be corrected.

*Reviewer #3 (Recommendations for the authors):*

1) CD138 expression alone is not necessarily sufficient to identify plasma cells. To determine the relationship between CD138+ cells induced in their in vitro system and in vivo plasma cells, the authors should perform a principal component analysis comparing the gene expression profile obtained from their RNA-seq analysis of CD138+B220lo cells with publicly available gene expression data from plasma cells. The authors could also show the percentage of CD138+B220lo cells that express key markers of plasma cell differentiation such as Blimp1.

2) The authors used mixed bone marrow chimera approaches to determine that YTHDF2 acts in a B cell intrinsic manner. However, these results do not indicate the stage of B cell differentiation where YTHDF2 acts to regulate plasma cell differentiation. Crossing the Ythdf2f/f mice to mice expressing Cre recombinase specifically in germinal center or plasma cells, such as the Jchain Cre, could clarify this issue. Alternatively, the authors could generate chimeras where they transduce Ythdf2f/f bone marrow with a tamoxifen inducible Cre to allow them to specifically ablate Ythdf2 after B cell activation has already occurred.

3) It is also unclear whether YTHDF2 is regulating migration to the bone marrow or maintenance within the bone marrow. A kinetic assessment of the frequency of bone marrow plasma cells in cKO mice could clarify this question.

4) The authors should include the number of times each experiment was repeated, and the total number of mice used in the figure legend. Based on the reporting form it seems that some experiments were only done once with 3-4 mice in the control group. Key experiments should be repeated at least twice with 3-6 mice per group to verify the reproducibility of their results.

---

## [Author Response]

Reviewer #1 (Recommendations for the authors):The authors show an important role of an RNA-binding protein (RBP), YTHDF2 in the accumulation of plasma cells. In addition, by a CRISPR/Cas9 knockout screening of RBPs, the authors suggest that some RBPs are involved in plasma cell differentiation. The roles of RBPs in a lymphocyte differentiation system are very interesting, but some technical problems should be solved before publication.1. The discrepancy of Figure 2C and 3D is confusing. Does YTHDR2 affect the formation of germinal centre B cells? If the authors just claim that GC can be formed, to support the cytometric data, histological analyses should be performed.

The measurements shown in Figures 2C and 3D of the original manuscript are of B cells with the GC phenotype at day 21 after immunisation. At this time, when the GC reaction is waning no conclusions can be drawn about the formation or function of germinal centres. Thus we have removed these data from the revised submission.

The paper by Grenov et al., is now published, in Figure 5C of that paper it shows a two-fold reduction in the frequency of GC B cells 14 days after administration of NP-KLH. In addition, it shows YTHDF2 is not required for the *Myc*-dependent gene expression programme associated with the selection of high-affinity B cell clones. These published data are consistent with our finding that YTHDF2 has no detectable effect on the GC cell numbers at day 7 in competitive chimeras, and a very small effect on the GC cellularity at day 21 in mMT chimeras. We include new data (Figure 2B) which shows no major effect on affinity maturation of the antibody response at day 21 in mMT chimeras.

The data of Grenov et al., and our findings, using independent systems and measurements suggest that any defect in the GC that does exist must be rather small in magnitude. The suggested histological analysis would require generating a new set of radiation chimeras in mMT mice, which we are unable to do. Histology may distinguish if the number or size of GCs is affected by the absence of YTHDF2, but we have no reason to think this is likely to be the case. Histology, in our experience, is not more sensitive than flow cytometry at detecting small differences in GC size.

2. Was the detection of NIP/IgG1 in CD138+ plasma cells performed by intracellular staining? Since most NP-specific IgG1-secreting plasma cells lose surface Igs, the detection may be underestimated. Alternatively, an ELISPOT assay can be also utilized.

We used intracellular staining to identify NIP and IgG1 expression in CD138+ CD267+ plasma cells in all experiments. We have edited the figure legends and methods [page 17, line 21] to make this clear.

3. The identification of YTHDF2-related genes in B/plasma cells should be further assessed. The protein expression of CXCR4, S1PR, integrin alpha4, etc. by flow cytometry will be supportive to Figure 4, even if it makes no difference.

We include new data (Figure 2—figure supplement 1B) that shows no difference in CXCR4 expression in CD138+ CD267+ cells between genotypes. We did not analyse S1PR or integrin α 4 at the protein level.

4. Grenov et al., (bioRxiv, 2020) show that B cell-specific METTL3-deficient mice fail to generate GC B cells and plasma cells. As METTL3 is closed to YTHDF2, the similarities and differences in the two papers should be discussed.

We agree and have added such discussion to our manuscript [page 10 line 26].

Reviewer #2 (Recommendations for the authors):Data presented in Figure 1C, 1F, 1G, Figure 2 and Figure 3 should be confirmed in at least 2 independent experiments performed on independent days with appropriate group sizes. The data presented here appears to be from one experimental cohort.

We have edited the manuscript to clarify that the experiments in Figure 1C, 1F, 1G and Figure 2 were all conducted on at least two independent occasions (this is included in the figure legends). In each experiment, data points are individual mice which are considered biological replicates. We have only conducted the measurements made in Figure 3 (competitive chimeras) in a single experiment with the indicated number of biological replicates. We have revised the text to clarify that the µMT and competitive chimera experiments are two independent experimental approaches to address the role of YTHDF2 in plasma cell differentiation [page 8, line 21]. Given that the competitive chimeras were generated as confirmatory support of the observations made with two independent cohorts of µMT chimeras and that the results from both systems are consistent with each other further repetition will not change the conclusion that YTHDF2 is required for accumulation of bone marrow plasma cells. We have clarified this in the transparent reporting file.

For the YTHDF2 knockout experiments did the authors also measure antibody titers to the high valency NP molecules? The data showing a small but significant drop in antibody titers to low-valency NP and if high-valency titers are normal this would suggests affinity maturation maybe affected in the absence of YTHDF2. This experiment would add support to the authors conclusion that the germinal center is not affected in the absence of YTHDF2. Additionally, ELISpot assays should be used as an additional method for enumerating NP-specific plasma cells in the spleens and bone marrow since the numbers that are counted by flow cytometry are so rare.

We measured antibody titres reactive with high valency NP and observed no difference between *Ythdf2* control and knockout µMT chimeras at day21 after NP-KLH immunisation. We have added new data to Figure 2B to show the ratio of high and low valency NP-binding IgG1 and conclude that the affinity maturation of the antibody response is not obviously diminished by the absence of YTHDF2 in B cells (we elaborate on this conclusion in response to reviewer 1 comment 1). Lack of correlation between antibody secreting cell numbers and serum antibody titers has been previously reported Takahashi et al., 1998; Holl et al., 2011, and has been previously observed in our group (Saveliev et al., Ref 7). Unfortunately, we do not have ELISPOT data, but our flow cytometry assays for detecting NP-binding cells as CD138+ CD267+ IRF4+ are robust and quantitative.

https://www.ncbi.nlm.nih.gov/pmc/articles/PMC2212188/

https://www.ncbi.nlm.nih.gov/pmc/articles/PMC3771081/

The statistical cutoffs and other metrics (i.e., z-score or fold change) used to define sgRNA enrichment should be clearly stated in the methods section, manuscript text, and/or figure legends.

We did not use a statistical cutoff for the enrichment of individual sgRNAs, instead we used a statistical cutoff for the combined enrichment of all sgRNAs targeting individual genes. The statistical cutoff for such a gene level enrichment was set at a Z-score of greater than or equal to +/- 2, and an FDR adjusted p-value of less than 0.05. We have clarified this in the revised version of the manuscript [page 15, line 16].

Details and steps for how the CRISPR/Cas9 screen, RNA-seq analysis, and eCLIP data was processed were included in the reporting summary but should also be included in the appropriate section of the manuscript.

We have revised the manuscript to include the relevant information from the reporting summary [page15].

Multiple typos were found throughout the manuscript that should be corrected.

We have carefully proofread our manuscript prior to resubmission and apologise for any typographical errors.

Reviewer #3 (Recommendations for the authors):1) CD138 expression alone is not necessarily sufficient to identify plasma cells. To determine the relationship between CD138+ cells induced in their in vitro system and in vivo plasma cells, the authors should perform a principal component analysis comparing the gene expression profile obtained from their RNA-seq analysis of CD138+B220lo cells with publicly available gene expression data from plasma cells. The authors could also show the percentage of CD138+B220lo cells that express key markers of plasma cell differentiation such as Blimp1.

In our experience, the technical differences between RNA-seq libraries prepared at different times and in different labs contribute a substantial amount of variability to the data, making direct comparisons very difficult. Thus, we believe a principle component analysis integrating our data with published plasma cell gene expression data would likely be very difficult to interpret. We include new data (Figure 1—figure supplement 1G) that shows the CD138+B220lo cells present in our cultures are also CD267+, IRF4+ and PRDM1+. These additional markers underpin our belief that the CD138+ B220lo cells have essential features of plasma cells. Moreover, the control genes in the screen *Bach2*, *Prdm1* and *Irf4* also enrich in a manner consistent with their expected impact on the differentiation of plasma cells, further substantiating the utility of the in vitro differentiation system for such a screen.

2) The authors used mixed bone marrow chimera approaches to determine that YTHDF2 acts in a B cell intrinsic manner. However, these results do not indicate the stage of B cell differentiation where YTHDF2 acts to regulate plasma cell differentiation. Crossing the Ythdf2f/f mice to mice expressing Cre recombinase specifically in germinal center or plasma cells, such as the Jchain Cre, could clarify this issue. Alternatively, the authors could generate chimeras where they transduce Ythdf2f/f bone marrow with a tamoxifen inducible Cre to allow them to specifically ablate Ythdf2 after B cell activation has already occurred.

We have demonstrated that *Ythdf2* deficiency does not impact the reconstitution of the mature B cell compartment. We have more clearly stated the limitations of our current work [page 10, line 17 and page 11 line 1] to acknowledge the uncertainty around the precise stage, or stages that *Ythdf2* acts in plasma cell differentiation and survival. We have revised the title of our manuscript to reflect this important point raised by the reviewer. A detailed mechanistic study using additional Cre mouse models is not possible to perform as we have not generated these mice. It would take us an additional 2 years to complete this revision and we believe this would be beyond the scope of this brief report.

3) It is also unclear whether YTHDF2 is regulating migration to the bone marrow or maintenance within the bone marrow. A kinetic assessment of the frequency of bone marrow plasma cells in cKO mice could clarify this question.

We agree that our data do not address the possibility that YTHDF2 has a role in the migration or survival of plasma cells and have edited the manuscript to ensure that this limitation is clear [page 10, line 17]. Given that both of our chimeric models indicate a reduction in the accumulation of antibody-secreting cells in the spleen at day 21 after immunisation we believe it is reasonable to suggest that YTHDF2 does not solely impact the accumulation of plasma cells through a role in migration. In light of Einstein et al., 2021, which describes a role for YTHDF2 in limiting endoplasmic reticulum stress-induced apoptosis https://pubmed.ncbi.nlm.nih.gov/34216543/, it is appealing to hypothesise that YTHDF2 also limits apoptosis due to ER stress in plasma cells. We have discussed this possibility in the revised version of the manuscript [page 10, line 24].

4) The authors should include the number of times each experiment was repeated, and the total number of mice used in the figure legend. Based on the reporting form it seems that some experiments were only done once with 3-4 mice in the control group. Key experiments should be repeated at least twice with 3-6 mice per group to verify the reproducibility of their results.

We apologise for omitting this information and causing ambiguity. We have updated the manuscript to include this [page 8 line 9, page 8 line 24 and figure legends]. Our competitive chimera experiment consisted of only 3 control mice and was conducted once. However, as this experiment was primarily undertaken to support the findings made with multiple cohorts of µMT chimeras and gave a consistent result we do not believe repeating it will add significant value to the manuscript. Therefore, we have taken care to indicate these limitations [page 8 line 24].

Furthermore, we include new data that shows a comparison between the absolute number of antigen specific antibody secreting cells that arise from WT CD45.1+ internal controls and *Ythdf2* deficient CD45.2+ cells in our competitive chimeras in N=6 mice (Figure 3—figure supplement 1C). These data enable the clear interpretation that YTHDF2-deficient plasma cells show defective accumulation in the bone marrow.